# Some Implicational Semilinear Gaggle Logics: (Dual) Residuated-Connected Logics

## Eunsuk Yang

Center for Humanities & Social Sciences, Department of Philosophy, Institute of Critical Thinking and Writing, Jeonbuk National University, Rm 417, Jeonju 54896, Korea; eunsyang@jbnu.ac.kr

**Abstract:** Implicational partial Galois logics and some of their *semilinear* extensions, such as semilinear extensions satisfying abstract Galois and dual Galois connection properties, have been introduced together with their relational semantics. However, similar extensions satisfying residuated, dual residuated connection properties have not. This paper fills the gaps by introducing those *semilinear* extensions and their relational semantics. To this end, the class of implicational (dual) residuated-connected prelinear gaggle logics is defined and it is verified that these logics are *semilinear*. In particular, associated with the contribution of this work, we note the following two: One is that implications can be introduced by *residuated connection* in semilinear logics. This shows that the residuated, dual residuated connection properties are important and so need to be investigated in semilinear logics. The other is that *set-theoretic* relational semantics can be provided for semilinear logics. Semilinear logics have been dealt with extensively in algebraic context, whereas they have not yet been performed in the set-theoretic one.

**Keywords:** fuzzy logics; (dual) residuated connection; semilinear logic; gaggles; Routley–Meyer-style semantics

## 1. Introduction

There have been many studies of fuzzy implications and inferences in theories and their applications, such as mathematical fuzzy logic, approximate reasoning, fuzzy control, image processing, data analysis, energy, healthcare, and transport (see [1–11] for recent works). This work is a contribution in mathematical fuzzy logic. As is well known, in order for a logic L to be *fuzzy*, it has to have truth degrees in evaluations and so the elements of a carrier set need to be at least *linearly ordered* (see Remark 2 below).

One of important trends in mathematical fuzzy logic is to introduce fuzzy logic systems with more general structures. Regarding this, Cintula [12] first introduced weakly implicative fuzzy logics as a subclass of weakly implicative logics (henceforth *WILs* for simplicity). After several years, he and Noguera [13] renamed them as weakly implicative *semilinear* logics (*WISLs* for simplicity) since the term "fuzzy" has many conflicting meanings. According to them [13,14], if a logic L is complete on linearly ordered algebraic models, then L is *semilinear*. For the detailed reasons for them to take the term "semilinear" in place of "fuzzy," see [13–15]. Hence, as the WILs were complete on the linearly ordered matrices, Cintula–Noguera [13] introduced WISLs. These facts show that in an algebraic context, they considered the term "semilinear."

After Dunn [16] first introduced *gaggles* to provide underlying algebraic structures for non-classical logics in general, he [17] soon generalized this concept to *partial* gaggles and *tonoids* so as to include more general structures, such as lattices and partially ordered sets. Since then, gaggle-related logics and relational semantics have been introduced (see [18–20] for recent works). In particular, Yang and Dunn [21] recently introduced the class of implicational tonoid logics and their Routley–Meyer-style relational semantics (RM-style semantics for simplicity) as well as algebraic semantics. Regarding this, note

the following two facts. One is that these logics are introduced as logics combining the class of tonoids introduced by Dunn [17] and the class of WILs introduced by Cintula [12]. Implicational tonoid logics are WILs with tonic properties. Because each component of a connective is isotone or antitone, Yang–Dunn introduced those logics as a subclass of WILs. The other is that both algebraic and set-theoretic semantics are provided for those logics. While algebraic semantics were only provided for WILs in [12], both set-theoretic and algebraic semantics were provided for implicational tonoid logics in [21]. For more detailed reasons to introduce implicational tonoid logics, see [21].

Yang and Dunn [22] further extended to implicational partial Galois logics as a specification of implicational tonoid logics and their related RM-style semantics. In particular, Yang [23] addressed *semilinear* extensions of implicational tonoid logics and some of the implicational partial Galois logics. Notice that, although Yang gave a new notion of the term "semilinear" to be applied in both algebraic and set-theoretic contexts, he dealt with semilinear extensions of some of the implicational partial Galois logics but not all of them. The implicational partial Galois logics include implicational tonoid logics satisfying either a (dual) Galois connection, (dual) residuated connection, or (dual) residuation. Among them, he just studied the semilinear extensions of implicational tonoid logics satisfying the (dual) Galois connection.

This series of facts gives rise to the following questions.

Q1. Can any semilinear extension of other implicational partial Galois logics be introduced?
Q2. Can any RM-style semantics be established for such extensions?

Here, as an affirmative answer to the questions, we address a semilinear subclass of the implicational tonoid logics satisfying the (dual) residuated connection and their RM-style semantics. The more exact reasons to do this are as follows. Some of them are already mentioned in [23]. First, the reason to investigate *semilinear* logics satisfying *tonicity*: As the results in [21] show, the tonic types of connectives in the algebraic semantics do not play a decisive role in completeness proof, whereas those types in the RM-style semantics do. See Remarks 2.11, 3.10, and 4.16 in [21]. It means that the tonicity property is of importance to relational semantics. However, this property has not yet been extensively studied in relational semantics for semilinear extensions of the implicational partial Galois logics. Thus, one needs to deal with tonicity more in the context of these extensions.

Second, the reason to investigate implicational tonoid semilinear logics satisfying generalized Galois properties: As mentioned above, Dunn [16] first introduced *gaggles*, the acronym of *generalized Galois logics*. Tonoids are a generalization of them obtained by dropping abstract Galois properties [17]. As is well known, substructural (semilinear) logics satisfy the Galois connection property. For instance, two negations $\sim, -$ and two implications $\rightarrow, \leftarrow$ are *Galois connected* in the basic substructural logic **GL** as follows:

$(GC1)$ $\beta \vdash_{\mathbf{GL}} \sim \alpha$ if and only if $\alpha \vdash_{\mathbf{GL}} -\beta$;

$(GC2)$ $\beta \vdash_{\mathbf{GL}} \alpha \rightarrow \gamma$ if and only if $\alpha \vdash_{\mathbf{GL}} \gamma \leftarrow \beta$.

This shows that to study implicational tonoid semilinear logics satisfying abstract Galois properties would be an interesting subject to researchers studying the Galois connection property and its generalizations.

Third, the reason to investigate implicational tonoid semilinear logics satisfying abstract *residuated, dual residuated connection properties*: Substructural (semilinear) logics further satisfy the residuated connection property. For instance, the pairs of intensional conjunction & and two implications $\rightarrow, \leftarrow$ $(\&, \rightarrow)$ and $(\&, \leftarrow)$ are *residuated connected* in **GL** as follows:

$(RC1)$ $\beta \& \alpha \vdash_{\mathbf{GL}} \gamma$ if and only if $\alpha \vdash_{\mathbf{GL}} \beta \rightarrow \gamma$;

$(RC2)$ $\beta \& \alpha \vdash_{\mathbf{GL}} \gamma$ if and only if $\beta \vdash_{\mathbf{GL}} \gamma \leftarrow \alpha$.

One important fact is that these implications can be introduced by this residuation from the conjunction in a semilinear logic. Algebraically, this can be considered as follows.

Given a groupoid operation $\circ$, its corresponding left and right division operations $\backslash$, $/$ can be obtained by residuation. For instance, $\backslash$ determined by $\circ$ can be defined as $a \backslash c := sup\{b : a \circ b \leq c\}$. In substructural logics, *residuated* lattice-ordered groupoids form their basic algebraic structures [24–26]. In this sense, to study the abstract residuated connection property is very important in semilinear logics and similarly for its dual property. Note that Yang [23] introduced semilinear extensions of implicational tonoid logics satisfying an abstract (dual) Galois connection property but not a (dual) residuated connection property.

Fourth, the reason to establish *set-theoretic* semantics for semilinear logics: Such semantics have been introduced very little, while algebraic semantics have been introduced extensively. For instance, the semantics for basic substructural semilinear logics recently introduced in [13,15,27–31] are all algebraic. However, many researchers in the tradition of philosophical logic would be interested in and familiar with set-theoretic semantics, such as possible worlds. Thus, this study would provide to them a familiar way to understand semilinear logics.

Fifth, the reason to investigate *RM-style* semantics in *labeled* language: After Routley and Meyer [32–34] first introduced ternary relational semantics, the so-called Routley–Meyer (*RM* for short) semantics, for relevance logics, these semantics have been used as representative relational semantics for substructural logics [35,36]. Although Yang [37] recently introduced semantics with the title "set-theoretic RM-style semantics" for the semilinear logic **MTL** (Monoidal t-norm logic), these semantics are different from the RM semantics. Because, while the frames of the former semantics have the same structures as algebraic semantics, the frames of the latter semantics do not. Note that Yang [23] very recently introduced RM-style set-theoretic semantics for implicational tonoid semilinear logics as an *n*-ary *generalization* of the ternary RM semantics. These semantics are slightly different from the semantics introduced by Bimbó and Dunn in [35]. The former semantics consider both the *labeled* and tonic types of a connective, whereas the latter ones deal with just the tonic type. By introducing the labeled types of connectives, we can distinguish between one property and its dual one, such as an abstract residuated connection and its dual connection (see Remark 1). Hence, this study would provide more general set-theoretic RM semantics with both labeling and tonicity maps for semilinear logics in the *labeled* language.

The more detailed organization for the work is as follows. In Section 2, implicational (dual) residuated-connected prelinear gaggle logics (for brevity, *I(D)RCPLGLs*) are introduced as implicational tonoid prelinear logics with an abstract (dual) residuated connection property. In Section 3, it is shown that these logics are semilinear in an algebraic sense, introduced by Cintula–Noguera. In Section 4, RM-style semantics are introduced for finitary those logics together with their soundness and completeness. Finally, in Section 5, a generalization of the term "semilinear" introduced by Yang [23] is recalled and it is shown that the I(D)RCPLGLs are semilinear in this sense. This generalized term can be applied to both algebraic and set-theoretic models.

For convenience, we finally introduce lists of some basic notations and abbreviations as Tables 1 and 2, respectively.

**Table 1.** Basic notations.

| Notions | Symbols/Letters |
| :---: | :---: |
| connectives | $*, *_1, *_2$ |
| sentences | $\alpha, \beta, \gamma, \delta, A, B$ |
| sets of sentences | $\Gamma, \Lambda, \Sigma$ |
| tonic types | $\pm, +, -$ |
| labeled types | $\Box, \Diamond$ |
| nodes | $u, v, w, d, e, k$ |

**Table 2.** Basic abbreviations.

| Abbreviations | Full Names |
|---|---|
| WIL(s) | weakly implicative logic(s) |
| WISL(s) | weakly implicative semilinear logic(s) |
| RM(-style) | Routley–Meyer(-style) |
| I(D)RCPGL(s) | implicational (dual) residuated-connected partial gaggle logic(s) |
| I(D)RCPLGL(s) | implicational (dual) residuated-connected prelinear gaggle logic(s) |
| T-RM$_\Rightarrow$ frame | implicational tonoid RM frame |
| (d)rcpG-RM$_\Rightarrow$ frame | implicational (dual) residuated-connected partial gaggle RM frame |

## 2. Preliminaries: I(D)RCPLGLs

In this section, the class of I(D)RCPLGLs is introduced as implicational tonoid prelinear logics satisfying the abstract (dual) residuated connection property (see [21,22]). A language $\mathcal{L}$ is defined as usual, i.e., as a countable propositional language equipped with $S$ (the set of sentences) built inductively from $AS$ (a set of atomic sentences) and $\mathbf{C}$ (a set of connectives) with the *arity* map $\mathbf{ar} : \mathbf{C} \to N$. A part of $\mathcal{L}$, where $\star \in \mathbf{C}$ and $\mathbf{ar}(\star) = n$, is denoted by $(\star, n)$. An $\mathcal{L}$-*substitution* is a map $sst : \mathbf{S}_\mathcal{L} \to \mathbf{S}_\mathcal{L}$ satisfying that $sst(\star(\alpha_1, \ldots, \alpha_n)) = \star(sst(\alpha_1), \ldots, sst(\alpha_n))$.

Henceforth, $\mathcal{L}$ is fixed as a countable propositional language. A *consecution* relation (briefly consecution) in $\mathcal{L}$ is a relation $\Lambda \vdash_R \alpha$, where a pair $\langle \Lambda, \alpha \rangle \in R$ and $\Lambda \cup \{R\} \subseteq \mathbf{S}_\mathcal{L}$; a *logic L* in $\mathcal{L}$ is a subclass of all consecutions subject to: *(i)* $\alpha \in \Lambda$ entails $\Lambda \vdash_L \alpha$; *(ii)* $\Lambda \vdash_L \alpha$ and $\Sigma \vdash_L \beta$ for each $\beta \in \Lambda$ entail $\Sigma \vdash_L \alpha$ (*Cut*); *(iii)* $\Lambda \vdash_L \alpha$ entails for any $\mathcal{L}$-substitution $sst$, $sst(\Lambda) \vdash_L sst(\alpha)$; a *theory* of a logic $L$ is a set of sentences satisfying that $\Lambda \vdash_L \alpha$ entails $\alpha \in \Lambda$.

A tonicity map $\mathbf{tt}$ is defined here as a function mapping each connective $\star$ such that $\mathbf{ar}(\star) = n > 0$ to its tonic type $\mathbf{tt}(\star) = (t_1, \cdots, t_n)$, where each $t_i$ is antitone $(-)$ or isotone $(+)$. A *tonic language* is henceforth introduced as $\mathcal{L}$ with the function $\mathbf{tt}$. If a tonic language $\mathcal{L}$ has a binary connective $\Rightarrow$ '$\Rightarrow$' is an abstract implication connective in the sense that the connectives $\to$ and $\leftarrow$ above are its concrete examples, where $(\Rightarrow, 2) \in \mathcal{L}$ and $\mathbf{tt}(\Rightarrow) = (-, +)$, it is called *implicational*; given an $n$-ary connective $\star$, $\star^n(\vec{\alpha}, \beta_i)$ denotes the application of $\star$ to $n$ arguments, where $\vec{\alpha}, \beta \in \mathbf{AS}$ are the sequence of arguments of $\star$ except its $i$-th one and its $i$-th argument, respectively.

**Definition 1.** *Let L be a logic with $(\Rightarrow, 2)$ in $\mathcal{L}$.*

*(i)* (*Implicational tonoid logic and linear theory* [21]) *L is said to be an implicational tonoid logic in case it has:*

*(transitivity, T)* $\beta \Rightarrow \gamma, \alpha \Rightarrow \beta \vdash_L \alpha \Rightarrow \gamma$.

*(modus ponens, MP)* $\alpha, \alpha \Rightarrow \beta \vdash_L \beta$.

*(reflexivity, R)* $\vdash_L \alpha \Rightarrow \alpha$.

*(tonicity, Ton$_\star^i$)* *For an arbitrary connective $\star \in \mathcal{L}$ such that $\mathbf{ar}(\star) = n > 0$ and each $i \leq n$,*

- *$\mathbf{tt}(\star)(i) = +$ entails $\alpha \Rightarrow \beta \vdash_L \star^n(\vec{\gamma}, \alpha_i) \Rightarrow \star^n(\vec{\gamma}, \beta_i)$, and*

- *$\mathbf{tt}(\star)(i) = -$ entails $\alpha \Rightarrow \beta \vdash_L \star^n(\vec{\gamma}, \beta_i) \Rightarrow \star^n(\vec{\gamma}, \alpha_i)$.*

*A theory $\Lambda$ in L is said to be* linear *in case, for any pair of sentences $\alpha, \beta$, $\Lambda \vdash_L \alpha \Rightarrow \beta$ or $\Lambda \vdash_L \beta \Rightarrow \alpha$.*

*(ii)* (*(Finitary) implicational prelinear tonoid logic*) *An implicational tonoid logic L is said to be prelinear in case L has the Linear Extension Property (LEP): for each theory $\Lambda$ and each sentence $\gamma$ satisfying $\Lambda \nvdash_L \gamma$, one can construct a linear theory $\Lambda'$ such that $\Lambda \subseteq \Lambda'$ and*

$\Lambda' \nvdash_L \gamma$. *Given a set of sentences $\Lambda$ such that $\Lambda \vdash_L \alpha$, a pair $\langle \Lambda, \alpha \rangle \in L$ is said to be a finitary consecution in case $\Lambda$ is a finite set, and $L$ finitary in case all the consecutional conditions for $L$ are finite.*

Let $\neg, \to$, and $\wedge$ be the negation, implication, and disjunction connectives in classical logic CL. The following are simple examples of isotonicity and antitonicity in $(Ton^i_\star)$.

**Example 1.** *Let $\vdash_{CL} \alpha \to \beta$ for all sentences $\alpha, \beta$.*

(*i*)　　*Isotonicity :*

　　(1)　　$\vdash_{CL} (\gamma \wedge \alpha) \to (\gamma \wedge \beta); \vdash_{CL} (\alpha \wedge \gamma) \to (\beta \wedge \gamma)$.

　　(2)　　$\vdash_{CL} (\gamma \to \alpha) \to (\gamma \to \beta)$.

(*ii*)　　*Antitonicity:*

　　(1)　　$\vdash_{CL} \neg \beta \to \neg \alpha$.

　　(2)　　$\vdash_{CL} (\beta \to \gamma) \to (\alpha \to \gamma)$.

The examples show that $\mathbf{tt}(\wedge) = (+, +)$, $\mathbf{tt}(\to) = (-, +)$, and $\mathbf{tt}(\neg) = (-)$.

We introduce the I(D)RCPLGLs as a subclass of implicational (dual) residuated-connected partial gaggle logics (I(D)RCPGLs for simplicity) introduced in [22]. For this, we need to improve a tonic language into a labeled language. A function *lt* is called a labeling map if it maps every $(\star, n)$ to its labeled type $\mathbf{lt}(\star) \in \{\Box, \Diamond\}$. A tonic language $\mathcal{L}$ equipped with a labeling map $\mathbf{lt}$ is called here *labeled language*. For I(D)RCPLGLs, $\mathcal{L}$ is henceforth fixed as a labeled language.

**Definition 2.** *Let $L$ be an implicational tonoid logic, where $(\Rightarrow, 2) \in \mathcal{L}$ and $\mathbf{tt}(\Rightarrow) = (-, +)$.*

(*i*)　　*(I(D)RCPGL [22]) $L$ is said to be an IRCPGL if it further has: for each $(\star_1, n), (\star_2, n) \in \mathcal{L}$, where $\star_1, \star_2 \in \mathbf{C}$ and $\mathbf{ar}(\star_1), \mathbf{ar}(\star_2) = n > 0$, and for each $i \leq n$, the notation "$\alpha \dashv\vdash \beta$" is used as shorthand for $\alpha \vdash \beta$ and $\beta \vdash \alpha$.*

$$(\text{residuated connection, RC}) \ \star_1{}^n_\Diamond(\vec{\gamma}, \alpha_i) \Rightarrow \beta \dashv\vdash_L \alpha \Rightarrow \star_2{}^n_\Box(\vec{\gamma}, \beta_i),$$

*where*

　　(1)　　*$\star_1$ and $\star_2$ have the tonic types different from each other in an argument distinct from $i$ and, especially,*

　　(2)　　*each of them has the isotone tonic type in its $i$-th argument.*

*$L$ is called an IDRCPGL if it further has: for each $(\star_1, n), (\star_2, n) \in \mathcal{L}$, where $\star_1, \star_2 \in \mathbf{C}$ and $\mathbf{ar}(\star_1), \mathbf{ar}(\star_2) = n > 0$, and for each $i \leq n$,*

$$(\text{dual RC, DRC}) \ \beta \Rightarrow \star_1{}^n_\Box(\vec{\gamma}, \alpha_i) \dashv\vdash_L \star_2{}^n_\Diamond(\vec{\gamma}, \beta_i) \Rightarrow \alpha,$$

*where $\star_1$ and $\star_2$ have the same tonic types as in (RC).*

(*ii*)　　*(I(D)RCPLGL) An I(D)RCPGL $L$ is said to be prelinear in case it has the (LEP). These logics are called here implicational (dual) residuated-connected prelinear gaggle logics (I(D)RCPLGLs for simplicity).*

Let $\neg, \to$, and $\wedge$ be as above and define $\to^d$ and $\wedge^d$ as follows: $\alpha \to^d \beta := \alpha \wedge \sim \beta$ and $\alpha \wedge^d \beta := \sim \beta \to \alpha$. The following are similar simple examples of (*RC*) and (*DRC*) in CL.

**Example 2.** *Let $\alpha, \beta, \gamma$ be sentences in CL.*

$(RC)$ $\alpha \to (\beta \to \gamma) \dashv\vdash_{CL} (\beta \wedge \alpha) \to \gamma$.

$(DRC)$ $(\gamma \to^d \beta) \to \alpha \dashv\vdash_{CL} \gamma \to (\alpha \wedge^d \beta)$.

**Remark 1.** *As Example 2 shows, (RC) and (DRC) are different from each other. However, we cannot distinguish them if we drop the labeling map, i.e., if we consider the properties in a tonic language. Note that $\star_1, \star_2$ are arbitrary n-ary connectives and $\alpha, \beta$ are arbitrary sentences in Definition 2 and so there is no difference between (RC) and (DRC) if we drop the labeled types of connectives.*

### 3. Semilinearity I

It is shown that I(D)RCPLGLs are semilinear in the sense of Cintula–Noguera, which is provided in an algebraic context (see [13]). First, for a pair of formulas $\alpha, \beta$, by $\alpha \Leftrightarrow \beta$, we denote the set of formulas $\{\alpha \Rightarrow \beta, \beta \Rightarrow \alpha\}$. Then, $\Leftrightarrow$ has the role of bi-implication.

**Definition 3** ([12,13]). *Let L be a logic with $(\Rightarrow, 2)$ in a countable propositional language $\mathcal{L}$.*

(*i*) *(WIL) L is called a WIL if it has $(T)$, $(MP)$, $(R)$, and the following: for each $(\star, n)$, where $\star \in \mathbf{C}$ and $\mathbf{ar}(\star) = n > 0$, and each $i \leq n$,*

*(congruence, $Cg_\star^i$) $\alpha \Leftrightarrow \beta \vdash_L \star^n(\vec{\gamma}, \alpha_i) \Rightarrow \star^n(\vec{\gamma}, \beta_i)$.*

(*ii*) *(Weakly implicative semilinear logic) A WIL L is called semilinear in case it is complete over linearly ordered L-matrices.*

**Fact 1.**

(*i*) *([21]) Implicational tonoid logics are WILs.*

(*ii*) *([22]) I(D)RCPGLs are implicational tonoid logics.*

(*iii*) *([23]) Implicational tonoid semilinear logics are weakly implicative semilinear logics.*

**Fact 2** ([12,13]). *For a WIL L in a countable propositional language $\mathcal{L}$, the following are equivalent:*

(*i*) *L has the LEP.*
(*ii*) *L is a semilinear logic.*

Since the class of I(D)RCPLGLs is a subclass of I(D)RCPGLs, we have the following as a corollary of the above two facts.

**Corollary 1.** *I(D)RCPLGLs are semilinear.*

**Remark 2.** *The corollary means that I(D)RCPLGLs are complete on linearly ordered matrices. The intended semantics of a semilinear logic L has the real unit interval [0, 1] as its carrier set and the completeness on [0, 1] of L is called standard completeness. However, lots of semilinear logics do not have such intended semantics and instead have semantics based on linearly ordered algebras or matrices (see, e.g., [28,31]). This shows that the linear ordering of algebraic models is the minimal condition for a logic to be semilinear.*

### 4. RM-Style Relational Semantics

As relational semantics, this section provides *RM-style semantics* for finitary I(D)RCPLGLs. As in [21–23], by the notion of a labeled language, we distinguish between connectives valued by existential sentences and connectives valued by universal sentences. In this section, L is fixed as a finitary I(D)RCPLGL in a labeled language $\mathcal{L}$.

#### 4.1. Notations

Suppose that $(F, \leq, R, D)$ is a relational structure such that $(F, \leq)$ is a linearly ordered set, $R$ is an $n+1$-ary relation on $F$, and $D \subseteq F$. By $R(u_1, \cdots, u_n; D)$, henceforth, we mean

that one can construct $d \in D$ such that $R(u_1, \cdots, u_n; d)$; given a $(\star, n)$, where $\mathbf{ar}(\star) = n > 0$, by $R_\star$, we mean an $n+1$-ary relation $R$ having the same labeling and tonicity maps as $\star$. Henceforth, we further fix $\mathcal{L}$ as a labeled language having $(\Rightarrow, 2) \in \mathcal{L}$, where $\mathbf{lt}(\Rightarrow) = \Box$ and $\mathbf{tt}(\Rightarrow) = (-, +)$.

Let $\star^n(\vec{\alpha}, \beta_i)$ be a sentence with an $n$-ary connective $\star$. By $R_\star(\vec{u}, v_i; w)$, we denote its corresponding $n+1$-ary relation such that $\vec{u}, v, w$ are the sequence of nodes forcing $\vec{\alpha}$ and the nodes forcing $\beta$ and the sentence $\star^n(\vec{\alpha}, \beta_i)$ itself, respectively. By the semicolon ';' in $R_\star$, we distinguish the node corresponding to the whole sentence from the nodes corresponding to atomic sentences appearing in $\star$. If one of membership $\in$ and non-membership $\notin$, one of antitonicity $-$ and isotonicity $+$, and one of forcing $\Vdash$ and non-forcing $\nVdash$ do not need to be specified, the notations '⋔,' '$\pm$,' and '⦀⊢,' respectively, are used.

We use the indices '$\Diamond(\pm)-$,' '$\Diamond(\pm)+$,' '$\Box(\pm)-$' and '$\Box(\pm)+$' in a valuation of a $(\star, n) \in \mathcal{L}$, where the notation '$\Diamond(\pm)-$' means the $i$-th argument place of $\star$ is antitone, its other argument places are antitone or isotone, and its labeled type is $\Diamond$, and similarly for the other indices. In its corresponding $n+1$-ary relation $R_\star$, we also use the same indices so as to emphasize that the $R$ preserves the labeled and tonic types of $\star$. If one does not have to consider the tonic types of $\star$ and $R_\star$, not all the indices above, but the indices '$\Box$' and '$\Diamond$' are only used in them. Moreover, if one has to distinguish the antitone and isotone parts in $(\pm)$, we instead use the notation '$(+, -)$' and analogously for '$(-)$' and '$(+)$.'

### 4.2. RM-Style Semantics

We first introduce implicational linear (dual) residuated gaggle RM frames.

**Definition 4.**

(i)　(Implicational tonoid RM frames (T-RM$_\Rightarrow$ frames for simplicity) [21]) We say that a structure $\mathcal{F} = (F, \leq, R_\Rightarrow, \{R_\star\}; D)$ is a T-RM$_\Rightarrow$ frame in case $(F, \leq)$ is a partially ordered set, $D \subseteq F$, $R_\Rightarrow \subseteq F^3$, and $R_\star \subseteq F^{n+1}$ for every $n$-ary connective $\star$ subject to the below conditions:

$(p_\leq)$ $u \leq v$ if and only if for all $u, v \in F$, $R_\Rightarrow(u, v; D)$;

$(p_M)$ for all $u, v, w \in F$,

- $R_\Rightarrow(u, v; w)$ and $u' \leq u$ imply $R_\Rightarrow(u', v; w)$,
- $R_\Rightarrow(u, v; w)$ and $v \leq v'$ imply $R_\Rightarrow(u, v'; w)$,
- $R_\Rightarrow(u, v; w)$ and $w' \leq w$ imply $R_\Rightarrow(u, v; w')$;

$(p_{MP})$ for any $u \in F$, $R_\Rightarrow(u, u; u)$;

$(df1)$　$R_{\star_\Diamond \Rightarrow}((\vec{u}, v_i), w; e)$ if and only if there is $k$ such that $R_{\star_\Diamond}(\vec{u}, v_i; k)$ and $R_\Rightarrow(k, w; e)$;

$(p_{Ton^i_{\star_\Diamond +}})$　if $R_{\star_{\Diamond(\pm)+} \Rightarrow}((\vec{w}, u_i), e; D)$, there is $k$ such that $R_{\star_{\Diamond(\pm)+}}(\vec{w}, k_i; e)$ and $u \leq k$;

$(p_{Ton^i_{\star_\Diamond -}})$　if $R_{\star_{\Diamond(\pm)-} \Rightarrow}((\vec{w}, u_i), e; D)$, there is $k$ such that $R_{\star_{\Diamond(\pm)-}}(\vec{w}, k_i; e)$ and $k \leq u$;

$(df1')$　$R_{\star_\Box \Rightarrow}(w, (\vec{u}, v_i); e)$ if and only if there is $k$ such that $R_{\star_\Box}(\vec{u}, v_i; k)$ and $R_\Rightarrow(w, k; e)$;

$(p_{Ton^i_{\star_\Box +}})$　if $R_{\star_{\Box(\pm)+} \Rightarrow}(e, (\vec{w}, u_i); D)$, there is $k$ such that $R_{\star_{\Box(\pm)+}}(\vec{w}, k_i; e)$ and $k \leq u$;

$(p_{Ton^i_{\star_\Box -}})$　if $R_{\star_{\Box(\pm)-} \Rightarrow}(e, (\vec{w}, u_i); D)$, there is $k$ such that $R_{\star_{\Box(\pm)-}}(\vec{w}, k_i; e)$ and $u \leq k$.

(ii)　(Implicational (dual) residuated-connected partial gaggle RM frames ((d)rcpG-RM$_\Rightarrow$ frames for simplicity) [22]) We say that a T-RM$_\Rightarrow$ frame $\mathcal{F} = (F, \leq, R_\Rightarrow, \{R_{\star_1}, R_{\star_2}\}; D)$ is an rcpG-RM$_\Rightarrow$ frame in case $R_{\star_1}$ and $R_{\star_2}$ are subject to the below definition and postulate.

(*df2*)  For $\star \in \{\star_1, \star_2\}$, $R_{\star \Rightarrow}(u, (\vec{v}, w_i); e)$ *if and only if there is $k$ such that* $R_\star(\vec{v}, w_i; k)$ *and* $R_\Rightarrow(u, k; e)$.

($p_{RC}$)  $R_{\star_{1 \diamond (\pm)+} \Rightarrow}((\vec{w}, u_i), v; D)$ *if and only if* $R_{\star_{2 \square (\pm)+} \Rightarrow}(u, (\vec{w}, v_i); D)$.

*We say that a T-RM$_\Rightarrow$ frame $\mathcal{F} = (F, \leq, R_\Rightarrow, \{R_{\star_1}, R_{\star_2}\}; D)$ is a drcpG-RM$_\Rightarrow$ frame in case $R_{\star_1}$ and $R_{\star_2}$ are subject to the definition (df2) and the below postulate.*

($p_{DRC}$)  $R_{\star_{1 \square (\pm)+} \Rightarrow}(v, (\vec{w}, u_i); D)$ *if and only if* $R_{\star_{2 \diamond (\pm)+} \Rightarrow}((\vec{w}, v_i), u; D)$.

*By (d)rcpG-RM$_\Rightarrow$ frames, we denote ambiguously rcpG-RM$_\Rightarrow$ and drcpG-RM$_\Rightarrow$ frames together.*

(*iii*)  (*Linear (d)rcpG-RM$_\Rightarrow$ frames ((d)rcpG-RM$_\Rightarrow^\ell$ frames for simplicity)*) *A (d)rcpG-RM$_\Rightarrow$ frame is called a (d)rcpG-RM$_\Rightarrow^\ell$ frame if $(F, \leq)$ is a linearly ordered set.*

**Remark 3.**

1.  *For convenience, we do not use the label $\square$ in each $\Rightarrow$.*

2.  *The postulate ($p_{Ton_{\star_{\square +}}^i}$) means that for every $(\star, n)$, where $\mathbf{ar}(\star) = n > 0$, if $\mathbf{lt}(\star) = \square$ and for the given $i$, $\mathbf{tt}(\star)(i) = +$, then there exists $k \in F$ such that $R_\star(\vec{w}, k_i; e)$ (where $k_i = k$) and $k \leq u$, and analogously for the other postulates.*

A *valuation* on a (d)rcpG-RM$_\Rightarrow^\ell$ frame is a relation $\Vdash$ between nodes and sentences satisfying: For every atomic sentence $p$,

(*AHC*) $u \leq v$ and $u \Vdash p$ imply $v \Vdash p$.

For sentences $\alpha, \beta$,

($\Rightarrow$) $w \Vdash \alpha \Rightarrow \beta$ if and only if for each $u, v \in F$, if $R_\Rightarrow(u, v; w)$ and $u \Vdash \alpha$, then $v \Vdash \beta$.

Additionally, for sentences $\vec{\alpha}, \beta$,

($\star_{\diamond +}^n$) $w \Vdash \star_{\diamond (\pm)+}^n(\vec{\alpha}, \beta_i)$ if and only if there are $\vec{u}, v \in F$ such that $R_{\star_{\diamond (\pm)+}}(\vec{u}, v_i; w)$,
$\vec{u} \Vvdash \vec{\alpha}$, and $v \Vdash \beta$;

($\star_{\diamond -}^n$) $w \Vdash \star_{\diamond (\pm)-}^n(\vec{\alpha}, \beta_i)$ if and only if there are $\vec{u}, v \in F$ such that $R_{\star_{\diamond (\pm)-}}(\vec{u}, v_i; w)$,
$\vec{u} \Vvdash \vec{\alpha}$, and $v \not\Vdash \beta$;

($\star_{\square +}^n$) $w \Vdash \star_{\square (\pm)+}^n(\vec{\alpha}, \beta_i)$ if and only if for all $\vec{u}, v \in F$, if $\vec{u} \Vvdash \vec{\alpha}$ and $R_{\star_{\square (\pm)+}}(\vec{u}, v_i; w)$,
then $v \Vdash \beta$;

($\star_{\square -}^n$) $w \Vdash \star_{\square (\pm)-}^n(\vec{\alpha}, \beta_i)$ if and only if for all $\vec{u}, v \in F$, if $v \Vdash \beta$ and $R_{\star_{\square (\pm)-}}(\vec{u}, v_i; w)$,
then $\vec{u} \Vvdash \vec{\alpha}$.

Let $\Lambda$ and $\alpha$ be a linear theory and a sentence, respectively, in L. We say that a pair $(\mathcal{F}, \Vdash)$ such that $\mathcal{F}$ is a (d)rcpG-RM$_\Rightarrow^\ell$ frame and $\Vdash$ is a valuation over $\mathcal{F}$ is a *(d)rcpG-RM$_\Rightarrow^\ell$ model* of $\Lambda$ in case for all $\alpha \in \Lambda$, $u \Vdash \alpha$ for every $u \in D$ ($D \Vdash \alpha$ for simplicity). Since $\leq$ in (d)rcpG-RM$_\Rightarrow^\ell$ frames and models is linearly ordered, we henceforth say such frames and models as *linear frames* and *linear models*. By $Mod^\ell(\Lambda, \mathcal{F})$, we denote the set of (d)rcpG-RM$_\Rightarrow^\ell$ models of $\Lambda$. We say that $\alpha$ is a *semantic consequence* of $\Lambda$ over $\mathfrak{F}$, expressed by $\Lambda \models_{\mathfrak{F}} \alpha$, in case $Mod^\ell(\Lambda, \mathcal{F}) = Mod^\ell(\Lambda \cup \{\alpha\}, \mathcal{F})$ for all $\mathcal{F} \in \mathfrak{F}$; and $\mathcal{F}$ as an **L** *frame* in case L $\subseteq \models_{\{\mathcal{F}\}}$. By $MOD^\ell(L)$, we denote the set of linear **L** frames and write $\Lambda \models_{L^\ell} \alpha$ in place of $\Lambda \models_{MOD^\ell(L)} \alpha$.

*4.3. Soundness and Completeness*

Here, we first prove that L is sound.

**Proposition 1.** *(Soundness) Let $\Lambda$ and $\alpha$ be a linear theory in L and a sentence, respectively. $\Lambda \vdash_L \alpha$ entails $\Lambda \models_{L^\ell} \alpha$.*

**Proof.** See Proposition 4.8 in [21] for the rules $(T)$, $(MP)$, $(Ton^i_\star)$ and the axiom $(R)$, and see Proposition 4.8 in [22] for the rule $(RC)$. Here, we prove $(DRC)$. For this, we have to verify that $D \Vdash \beta \Rightarrow \star 1^n_{\square(\pm)+}(\vec{\gamma}, \alpha_i)$ if and only if $D \Vdash \star 2^n_{\Diamond(\pm)+}(\vec{\gamma}, \beta_i) \Rightarrow \alpha$.

$(\Longrightarrow)$ We assume $D \Vdash \beta \Rightarrow \star 1^n_{\square(\pm)+}(\vec{\gamma}, \alpha_i)$ and prove $D \Vdash \star 2^n_{\Diamond(\pm)+}(\vec{\gamma}, \beta_i) \Rightarrow \alpha$. To prove this, by the condition $(\Rightarrow)$, we further suppose that for any $k', u$, $R_\Rightarrow(k', u; D)$ and $k' \Vdash \star 2^n_{\Diamond(\pm)+}(\vec{\gamma}, \beta_i)$ and prove $u \Vdash \alpha$. Since $k' \Vdash \star 2^n_{\Diamond(\pm)+}(\vec{\gamma}, \beta_i)$, the condition $(\star^n_{\Diamond+})$ assures that one can construct $\vec{v}, e$ such that $R_{\star 2\Diamond(\pm)+}(\vec{v}, e_i; k')$, $\vec{v} \Vvdash \vec{\gamma}$ and $e \Vdash \beta$. We also obtain $R_{\star 1\square(\pm)+}(\vec{v}, u_i; k)$ and $R_\Rightarrow(e, k; D)$ for some $k$, using the postulate $(p_{DRC})$ and $(df2)$, and so $k \Vdash \star 1^n_{\square(\pm)+}(\vec{\gamma}, \alpha_i)$. Then, since $\vec{v} \Vvdash \vec{\gamma}$, we obtain that $u \Vdash \alpha$ by $(\star^n_{\square+})$.

$(\Longleftarrow)$ We assume $D \Vdash \star 2^n_{\Diamond(\pm)+}(\vec{\gamma}, \beta_i) \Rightarrow \alpha$ and prove $D \Vdash \beta \Rightarrow \star 1^n_{\square(\pm)+}(\vec{\gamma}, \alpha_i)$. To prove this, by the condition $(\Rightarrow)$, we further suppose that for any $k', u$, $R_\Rightarrow(u, k'; D)$ and $u \Vdash \beta$ and prove that $k' \Vdash \star 1^n_{\square(\pm)+}(\vec{\gamma}, \alpha_i)$. For this, we further assume that for any $\vec{v}, e$, $R_{\star 1\square(\pm)+}(\vec{v}, e_i; k')$ and $\vec{v} \Vvdash \vec{\gamma}$, and show that $e \Vdash \alpha$. Using the postulate $(p_{DRC})$ and $(df2)$, we can obtain that $R_{\star 2\Diamond(\pm)+}(\vec{v}, u_i; k)$ and $R_\Rightarrow(k, e; D)$ for some $k$, and thus $k \Vdash \star 2^n_{\Diamond(\pm)+}(\vec{\gamma}, \beta_i)$. Hence, using $R_\Rightarrow(k, e; D)$, we further have that $e \Vdash \alpha$ by $(\Rightarrow)$.

Notice finally that L can be sound over $MOD^\ell(L)$ since it is an I(D)RCPLGL in $\mathcal{L}$ and $\Lambda$ is a linear theory in L. □

This sentence ensures that (d)rcpG-RM$^\ell_\Rightarrow$ frames for L are linear **L** frames.

Now, we provide completeness. By an L-theory, we mean a theory $\Lambda$ closed under rules of L. By a regular L-theory, we mean an L-theory containing all of the theorems of L. Since we have no use for irregular theories, by an L-theory, we henceforth mean an L-theory containing all of the theorems of L.

We say that a canonical structure $\mathcal{F} = (F^{can}, \leq^{can}, R^{can}_\Rightarrow, \{R^{can}_\star : \star \in \{\star_1, \star_2\}\}; D^{can})$ is the *canonical (d)rcpG-RM$^\ell_\Rightarrow$ frame* determined by $\Lambda$, where $\Lambda$ is a linear theory of L such that $\Lambda \nvdash_L \alpha$, in case $F^{can}$ is the set of linear theories extending $\Lambda$ in L, $\leq^{can}$ is a subset relation $\subseteq$ restricted to $F^{can}$, $D^{can} = \{\Lambda\}$, and $R^{can}$'s are defined:

(a)　　$R^{can}_\Rightarrow(u, v; w)$ if and only if for all $\alpha, \beta$, $\alpha \Rightarrow \beta \in w$ and $\alpha \in u$ entail $\beta \in v$.

(b)　　For each $(\star, n) \in \mathcal{L}$ and each $i \leq n$,

　　(1)　$R^{can}_{\star_\Diamond(\pm)+}(\vec{u}, v_i; w)$ if and only if for all $\vec{\alpha}, \beta$, $\vec{\alpha} \Vvdash \vec{u}$ and $\beta \in v$ entail $\star^n(\vec{\alpha}, \beta_i) \in w$;

　　(2)　$R^{can}_{\star_\Diamond(\pm)-}(\vec{u}, v_i; w)$ if and only if for all $\vec{\alpha}, \beta$, $\vec{\alpha} \Vvdash \vec{u}$ and $\beta \notin v$ entail $\star^n(\vec{\alpha}, \beta_i) \in w$;

　　(3)　$R^{can}_{\star_\square(\pm)+}(\vec{u}, v_i; w)$ if and only if for all $\vec{\alpha}, \beta$, $\star^n(\vec{\alpha}, \beta_i) \in w$ and $\vec{\alpha} \Vvdash \vec{u}$ entail $\beta \in v$;

　　(4)　$R^{can}_{\star_\square(\pm)-}(\vec{u}, v_i; w)$ if and only if for all $\vec{\alpha}, \beta$, $\star^n(\vec{\alpha}, \beta_i) \in w$ and $\beta \in v$ entail $\vec{\alpha} \Vvdash \vec{u}$.

**Proposition 2.** *A (d)rcpG-RM$^\ell_\Rightarrow$ frame defined canonically is linearly ordered.*

**Proof.** See Proposition 2 in [23]. □

We then prove that the $\mathcal{F}$ is a (d)rcpG-RM$^\ell_\Rightarrow$ frame.

**Lemma 1.** *(Lemma 4.10, [21]) Let $\mathcal{F}$ be a canonical (d)rcpG-RM$^\ell_\Rightarrow$ frame and $\alpha, \beta$ be sentences. If $D^{can} \Vdash^{can} \alpha \Rightarrow \beta$, then for each $u \in F^{can}$, $u \Vdash^{can} \alpha$ entails $u \Vdash^{can} \beta$.*

**Lemma 2.** *The postulates for (d)rcpG-RM$_{\Rightarrow}^{\ell}$ frames are satisfied in the above relation $R^{can}$ defined canonically.*

**Proof.** We consider the postulate $(p_{DRC}^{can})$ since the other postulates are proved in Lemma 4.11 in [21] and Lemma 4.10 in [22].

For $(p_{DRC}^{can})$, one has to prove that:

$$R_{\star_{1\square(\pm)+}\Rightarrow}^{can}(v,(\vec{w},u_i);D^{can}) \text{ if and only if } R_{\star_{2\diamond(\pm)+}\Rightarrow}^{can}((\vec{w},v_i),u;D^{can}).$$

($\Longleftarrow$) We first suppose that $R_{\star_{2\diamond(\pm)+}\Rightarrow}^{can}((\vec{w},v_i),u;D^{can})$. We must prove $R_{\star_{1\square(\pm)+}\Rightarrow}^{can}(v,(\vec{w},u_i);D^{can})$, i.e., one can construct a linear theory $d$ in L such that $R_{\star_{1\square(\pm)+}}^{can}(\vec{w},u_i;d)$ and $R_{\Rightarrow}^{can}(v,d;D^{can})$. Let $d' =_{df} \{\star^n(\vec{\gamma},\alpha_i) : \exists \delta \in v\, (\Lambda \vdash_L \delta \Rightarrow \star^n(\vec{\gamma},\alpha_i))\}$. To prove that $d'$ is a linear theory in an IDRCPLGL L, we first deal with its rules.

($MP$): Assume that $A \Rightarrow B \in d'$ and $A \in d'$. One has to verify $B \in d'$. By the definition of $d'$, we instead suppose that one can construct $\delta, \delta' \in v$ such that $\Lambda \vdash_L \delta \Rightarrow (A \Rightarrow B)$ and $\Lambda \vdash_L \delta' \Rightarrow A$. Then, Lemma 1 assures $A, A \Rightarrow B \in v$. Thus, we obtain $B \in d'$ because $\Lambda \vdash_L B \Rightarrow B$.

The proof for $(T)$ and $(Ton_\star^i)$ is analogous.

($DRC$): We consider the case $\beta \Rightarrow \star_{1\square}^n(\vec{\gamma},\alpha_i) \vdash_L \star_{2\diamond}^n(\vec{\gamma},\beta_i) \Rightarrow \alpha$. Suppose that $\beta \Rightarrow \star_{1\square}^n(\vec{\gamma},\alpha_i) \in d'$. We need to verify that $\star_{2\diamond}^n(\vec{\gamma},\beta_i) \Rightarrow \alpha \in d'$. As above, we instead suppose that one can construct $\delta \in v$ such that $\Lambda \vdash_L \delta \Rightarrow (\beta \Rightarrow \star_{1\square}^n(\vec{\gamma},\alpha_i))$. Lemma 1 ensures that $\beta \Rightarrow \star_{1\square}^n(\vec{\gamma},\alpha_i) \in v$. Then, by $(DRC)$, we have $\star_{2\diamond}^n(\vec{\gamma},\beta_i) \Rightarrow \alpha \in v$. Thus, since $\Lambda \vdash_L (\star_{2\diamond}^n(\vec{\gamma},\beta_i) \Rightarrow \alpha) \Rightarrow (\star_{2\diamond}^n(\vec{\gamma},\beta_i) \Rightarrow \alpha)$, we obtain that $\star_{2\diamond}^n(\vec{\gamma},\beta_i) \Rightarrow \alpha \in d'$.

It is clear that the theory $d'$ is linear because $\Lambda \subseteq d'$.

Now, we consider $R_{\star_{1\square(\pm)+}}^{can}(\vec{w},u_i;d')$ and $R_{\Rightarrow}^{can}(v,d';D^{can})$. Assume that $\delta \in v$ and $\Lambda \vdash_L \delta \Rightarrow \star_1^n(\vec{\gamma},\alpha_i)$ for some $\delta$. The condition $(a)$ assures $R_{\Rightarrow}^{can}(v,d';D^{can})$ because $\star_1^n(\vec{\gamma},\alpha_i) \in d'$ by the definition of $d'$. For $R_{\star_{1\square(\pm)+}}^{can}(\vec{w},u_i;d')$, suppose $\vec{\gamma} \pitchfork \vec{w}$. We must prove that $\alpha \in u$. Note first that it can be supposed that one can construct a linear theory $k$ in L such that $R_{\star_{2\diamond(\pm)+}}^{can}(\vec{w},v_i;k)$ and $R_{\Rightarrow}^{can}(k,u;D^{can})$. We obtain $\star_2^n(\vec{\gamma},\delta_i) \in k$ using $(b)$ (1) and the suppositions. Hence, since $\delta \Rightarrow \star_1^n(\vec{\gamma},\alpha_i) \in \Lambda$ implies $\star_2^n(\vec{\gamma},\delta_i) \Rightarrow \alpha \in \Lambda$, we obtain $\alpha \in u$ by $R_{\Rightarrow}^{can}(k,u;D^{can})$ and $(a)$.

($\Longrightarrow$) Suppose that $R_{\star_{1\square(\pm)+}\Rightarrow}^{can}(v,(\vec{w},u_i);D^{can})$. We have to prove that $R_{\star_{2\diamond(\pm)+}\Rightarrow}^{can}((\vec{w},v_i),u;D^{can})$, i.e., one can construct a linear theory $d$ in L such that $R_{\Rightarrow}^{can}(d,u;D^{can})$ and $R_{\star_{2\diamond(\pm)+}}^{can}(\vec{w},v_i;d)$. Let $d' =_{df} \{E : \exists \vec{\gamma} \pitchfork \vec{w}, \beta \in v\, (\Lambda \vdash_L \star_2^n(\vec{\gamma},\beta_i) \Rightarrow E)\}$. As above, clearly, $d'$ is a linear theory in L. Suppose that there are $\vec{\gamma} \pitchfork \vec{w}$ and $\beta \in v$. Then, by the definition of $d'$, we have that $\star_2^n(\vec{\gamma},\beta_i) \in d'$ since $\star_2^n(\vec{\gamma},\beta_i) \Rightarrow \star_2^n(\vec{\gamma},\beta_i) \in \Lambda$. Hence, by $(b)$ (1), we obtain $R_{\star_{2\diamond(\pm)+}}^{can}(\vec{w},v_i;d')$. To verify $R_{\Rightarrow}^{can}(d',u;D^{can})$, suppose moreover that $\star_2^n(\vec{\gamma},\beta_i) \Rightarrow \alpha \in \Lambda$ and $\star_2^n(\vec{\gamma},\beta_i) \in d'$. We have to prove that $\alpha \in u$. Since $R_{\star_{1\square(\pm)+}\Rightarrow}^{can}(v,(\vec{w},u_i);D^{can})$, we further suppose that one can construct a linear theory $k$ in L such that $R_{\star_{1\square(\pm)+}}^{can}(\vec{w},u_i;k)$ and $R_{\Rightarrow}^{can}(v,k;D^{can})$. Since $\star_2^n(\vec{\gamma},\beta_i) \Rightarrow \alpha \in \Lambda$ implies $\beta \Rightarrow \star_1^n(\vec{\gamma},\alpha_i) \in \Lambda$, using $R_{\Rightarrow}^{can}(v,k;D^{can})$ and $(a)$, we have that $\star_1^n(\vec{\gamma},\alpha_i) \in k$; therefore, $\alpha \in u$ by $R_{\star_{1\square(\pm)+}}^{can}(\vec{w},u_i;k)$ and $(b)$ (3). $\square$

**Theorem 1.** *The structure $\mathcal{F} = (F^{can}, \leq^{can}, R_{\Rightarrow}^{can}, \{R_\star^{can} : \star \in \{\star_1, \star_2\}\}; D^{can})$, which is defined canonically, is a (d)rcpG-RM$_{\Rightarrow}^{\ell}$ frame.*

**Proof.** One is capable of obtaining the claim by the canonically defined structure and Lemma 2. $\square$

Define a canonical valuation for a finitary I(D)RCPLGL L as follows:

$$(V^{can})\, u \Vdash^{can} \gamma \text{ if and only if } \gamma \in u.$$

**Proposition 3.** *(Canonical Valuation, Lemma 4.13 [21]) The $\Vdash^{can}$ defined canonically is a valuation.*

By Proposition 3, one can ensure that the $(\mathcal{F}, \Vdash^{can})$ defined canonically is indeed a (d)rcpG-RM$^\ell_\Rightarrow$ model.

**Theorem 2.** (*Completeness*) *Let* $\Lambda$ *and* $\alpha$ *be a theory in L and a sentence, respectively.* $\Lambda \models_{L^\ell} \alpha$ *entails* $\Lambda \vdash_L \alpha$.

**Proof.** Assume contrapositively that $\Lambda \nvdash_L \alpha$. One is capable of constructing a linear theory $\Lambda' \supseteq \Lambda$ such that $\Lambda' \nvdash_L \alpha$. Then, one can have $\Lambda' \nVdash^{can} \alpha$ using ($V^{can}$). Hence, by Proposition 3, one can further obtain $\Lambda' \nmodels_{L^\ell} \alpha$ and thus $\Lambda \nmodels_{L^\ell} \alpha$. □

**Corollary 2.** *Let* $\Lambda$ *and* $\alpha$ *be a theory in L and a sentence, respectively.* $\Lambda \vdash_L \alpha$ *if and only if* $\Lambda \models_{L^\ell} \alpha$.

### 5. Semilinearity II

The notion "semilinearity" was generalized as a notion applied in both algebraic and set-theoretic contexts [23]. Here, we verify that finitary I(D)RCPLGLs are semilinear in this sense. We henceforth fix $\mathcal{L}$ as a labeled language.

**Definition 5.** (*Semilinear implication and semilinear logic, cf.* [23]) *Given an I(D)RCPGL L in* $\mathcal{L}$*, we say that the connective* $\Rightarrow$ *is a semilinear implication if the linear models defined by* $\Rightarrow$ *establish sound and complete semantics for L. We say that L is semilinear if L has a semilinear implication.*

**Proposition 4.**

(*i*)     *If L is a finitary I(D)RCPLGL in* $\mathcal{L}$*, then L is semilinear.*

(*ii*)    *If L is a finitary I(D)RCSGL in* $\mathcal{L}$*, then L is prelinear.*

**Proof.** (*i*) One can obtain the claim using Definition 5 and Corollary 2.
(*ii*) Suppose that L, $\Lambda$, and $\gamma$ are a finitary I(D)RCSGL, a linear theory in L, and a sentence, respectively. One is capable of saying that if $\Lambda \nvdash_L \gamma$, there is a (d)rcpG-RM$^\ell_\Rightarrow$ frame $D$ and a valuation $\Vdash$ such that $D \nVdash \gamma$, since L is complete. Assume $\Lambda \nvdash_L \gamma$ and take $\Sigma$ as $\Lambda \bigcup \{\alpha : D \Vdash \alpha\}$. Then, we have $\Lambda \subseteq \Sigma$ and $\Sigma \nvdash_L \gamma$. Consider $u, v$ such that $u \Vdash \alpha$ and $v \Vdash \beta$ for all $\alpha, \beta$ so as to verify that $\Sigma$ is linear. Since $\leq$ is a linear order, clearly, $u \leq v$ or $v \leq u$. Let $u \leq v$. By the condition ($\Rightarrow$), we can have $D \Vdash \alpha \Rightarrow \beta$. Analogously, we can have $D \Vdash \beta \Rightarrow \alpha$ in case $v \leq u$. Hence, we have $D \Vdash \alpha \Rightarrow \beta$ or $D \Vdash \beta \Rightarrow \alpha$ and thus $\alpha \Rightarrow \beta \in \Sigma$ or $\beta \Rightarrow \alpha \in \Sigma$. □

**Corollary 3.** *Given a finitary I(D)RCPGL L in* $\mathcal{L}$*, the following are equivalent:*

(*i*)     *L is sound and complete over the class of all (d)rcpG-RM$^\ell_\Rightarrow$ frames.*

(*ii*)    *L is semilinear.*

(*iii*)   *L is prelinear.*

**Remark 4.** *The corollary ensures that finitary I(D)RCPGLs are complete on linearly ordered frames.*

### 6. Concluding Remarks

I(D)RCPLGLs were first defined and then dealt with as semilinear logics in the algebraic sense of Cintula–Noguera. Next, RM-style relational semantics were provided for those logics, and it was proved that finitary those logics are sound and complete on the semantics. Finally, it was shown that these finitary logics are semilinear in the set-theoretic sense of Yang.

Implicational semilinear logics have been introduced as a generalization of weakly implicative semilinear logics by Cintula and Noguera [38,39]. According to this generalization, weakly implicative semilinear logics form a subclass of implicational semilinear

logics. Let $\mathcal{I(D)RCSGL}$, $\mathcal{ITSL}$, $\mathcal{WISL}$, and $\mathcal{ISL}$ be the classes of implicational (dual) residuated-connected semilinear gaggle logics, implicational tonoid semilinear logics, weakly implicative semilinear logics, and implicational semilinear logics, respectively. The following subset relations hold between the classes.

$$\mathcal{I(D)RCSGL} \subset \mathcal{ITSL} \subset \mathcal{WISL} \subset \mathcal{ISL}.$$

Finally, note that here the models for finitary I(D)RCPLGLs are set-theoretic, while the models for weakly implicative and implicational semilinear logics in [14,38,39] are algebraic. Let weak p-implicational and weakly implicative matrices be matrices for p-implicational and WILs, respectively, and $\mathcal{FITSL}$ and $\mathcal{FI(D)RCSGL}$ denote the $\mathcal{ITSL}$ and $\mathcal{I(D)RCSGL}$ restricted to finitary logics, respectively. Table 3 summarizes the difference.

**Table 3.** Logics and Semantics.

| Logic | Structure | Semantic Method |
|---|---|---|
| $\mathcal{ISL}$ | linear weak p-implicational matrix | algebraic |
| $\mathcal{WISL}$ | linear weakly implicative matrix | algebraic |
| $\mathcal{FITSL}$ | T-RM$^{\ell}_{\Rightarrow}$ frame | relational |
| $\mathcal{FI(D)RCSGL}$ | (d)rcpG-RM$^{\ell}_{\Rightarrow}$ frame | relational |

Recall that I(D)RCPLGLs are logics with tonicity and the tonic and labeled properties of connectives are very important in providing relational semantics for finitary those logics. Since I(D)RCPLGLs are also implicational semilinear logics, to deal with finitary implicational semilinear logics with tonicity and their relational semantics is an interesting future work. Furthermore, since I(D)RCPLGLs are a specification of implicational tonoid semilinear logics, one may also consider a specification of I(D)RCPLGLs such as finitary implicational lattice-ordered prelinear logics and their RM-style semantics. This is another future work to be investigated.

**Funding:** This work was supported by the Ministry of Education of the Republic of Korea and the National Research Foundation of Korea (NRF-2019S1A5A2A01034874).

**Data Availability Statement:** Not applicable.

**Conflicts of Interest:** The author declares no conflict of interest.

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
