# Peer review of "Some Implicational Semilinear Gaggle Logics: (Dual) Residuated-Connected Logics"

_axioms, doi:10.3390/axioms11040183_

Round 1
Reviewer 1 Report
Dear Authors
All the parts of the manuscripts should be extended by giving detailed explanation. in current form there is no interest for the readers.
Author Response
To the reviewer #1,
Please see the attachment.
Thanks.

Reviewer 2 Report
In this manuscript, the authors present a fuzzy extension of a specific type of gaggle logic.
In my opinion, it is difficult to see the authors’ contributions in fuzzy logic theory. They provide a list of notations and definitions about new type of logics (languages, sentences and two semantics) and present them with several propositions, lemmas and two theorems about their peculiarities.
In my opinion, there is some discrepancy between classical fuzzy logic and proposed logical structures.
The practical implications of the authors' considerations have not been clarified (there are no practical examples/experimental part). Perhaps the authors could include an illustrative example on how to implement the semilinearity in some use cases in the manuscript.
Technical remarks:
In my opinion, the authors should emphasize the role of ‘fuzziness’ for implicational prelinear gaggle logics. The term “fuzzy logic” is missing in the keyword list and the manuscript body, which is not correct.
In “References”, more details about similar previous studies from the last five years should be included. In my opinion, there are too many self-citations (6 out of 28 references + 2 citations of a co-author).
Author Response
To the reviewer #2,
Please see the attachment.
Thanks.

Reviewer 3 Report
This paper fuzzy extensions of implicational (dual) residuated connected partial gaggle logics by defining the class of implicational (dual) residuated-connected pre-linear gaggle logics and verification of semilinear in the algebraic sense of CintulaNoguera.
Overall, the paper is interesting for the researcher’s community in the field of fuzzy theory. A detailed comment is as follows.
Comments:
- The title is a bit confusing; could you please revise it.
- The abstract is missing the main motivation and the problem statement. It should be revised with a clear motivation, problem statement, and objective.
- In the abstract Yang is mentioned, what is that? Is it the Yang method? If so there should be some name for the method, that should be used, instead.
- The introduction is tailored to the implication logic while lacking to present their application in various fields such as energy (DOI: 10.1109/ACCESS.2019.2917297), healthcare, transport (DOI: 10.1109/TITS.2022.3140461), etc. with vague information.
- In the preliminary section, there are several symbols such as ♣, and ♡ What are those? Also, there are lots of mathematical symbols and variables, some of which are even not defined in the text. Could you please provide a nomenclature list for the sake of the reader’s clarity?
- In Semi linearity I, why do you define α ⇔ β as {α ⇒ β, β ⇒ α}. Both have the same meaning. Isn’t it? Please explain.
- It is hard to understand the main contribution. Perhaps, I missed it, could you please clearly state what is the main contribution of this work?
- How to evaluate the effectiveness of the proposed method? There should be a comparative study. Unfortunately, I can’t figure out the efficiency of the proposed method. Could you please explain?
Author Response
To the reviewer #3,
Please see the attachment.
Thanks.

Round 2
Reviewer 1 Report
Dear Authors
thanks for the revised version, still I think the paper is not suitable for an SCI indexed journal.
Reviewer 2 Report
Technical remark:
Abstract: 'However, similar extensions satisfying residuated, dual residuated connection properties have not, although implicational partial gaggle logics satisfying them were already introduced.' - Please, edit this fragment.
The quality of axioms-1586202-v2 “Some implicational semilinear gaggle logics: (dual) residuated-connected logics” has been improved significantly. In my opinion, the manuscript meets the requirements of MDPI Axioms Journal.
My recommendation is “Accept as is”.
Author Response
To the reviewer #2,
I appreciate for your helpful comment, I have improved my manuscript following your comment. For this, please see the “=> part” below as my response to the comment.
Comments and Suggestions for Authors
Technical remark:
Abstract: 'However, similar extensions satisfying residuated, dual residuated connection properties have not, although implicational partial gaggle logics satisfying them were already introduced.' - Please, edit this fragment.
=> Following your comments, I deleted the part “although implicational partial gaggle logics satisfying them were already introduced” since it is unnecessary (see the yellow part in Abstract in p1).
The quality of axioms-1586202-v2 “Some implicational semilinear gaggle logics: (dual) residuated-connected logics” has been improved significantly. In my opinion, the manuscript meets the requirements of MDPI Axioms Journal.
My recommendation is “Accept as is”.

Reviewer 3 Report
I appreciate the author's efforts in improving the quality of the manuscript. The revised version of the manuscript is recommended for publication.
Author Response
Please see the attachment,
